# Synthetic protein interactions reveal a functional map of the cell

**Lisa K Berry, Guðjón Ólafsson, Elena Ledesma-Fernández[†], Peter H Thorpe***

Mitotic Control Laboratory, The Francis Crick Institute, Mill Hill Laboratory, London, United Kingdom

**Abstract** To understand the function of eukaryotic cells, it is critical to understand the role of protein-protein interactions and protein localization. Currently, we do not know the importance of global protein localization nor do we understand to what extent the cell is permissive for new protein associations – a key requirement for the evolution of new protein functions. To answer this question, we fused every protein in the yeast *Saccharomyces cerevisiae* with a partner from each of the major cellular compartments and quantitatively assessed the effects upon growth. This analysis reveals that cells have a remarkable and unanticipated tolerance for forced protein associations, even if these associations lead to a proportion of the protein moving compartments within the cell. Furthermore, the interactions that do perturb growth provide a functional map of spatial protein regulation, identifying key regulatory complexes for the normal homeostasis of eukaryotic cells.

**\*For correspondence:** peter. thorpe@crick.ac.uk

**Present address:** [†]MRC Laboratory of Molecular Cell Biology, University College London, London, United Kingdom

**Competing interests:** The authors declare that no competing interests exist.

## Introduction

Post-translational protein modifications such as phosphorylation or ubiquitylation often alter the affinity of one protein for other proteins or cellular components, which drive their movement within the cell (*Scott and Pawson, 2009*). Protein relocalization is critical for many cellular processes, including the asymmetric division of adult stem cells, which underlies metazoan development. The importance of protein localization is also highlighted by diseases ranging from cystic fibrosis to cancer that result, in part, from protein mislocalization (*Hung and Link, 2011*). The evolution of new modes of protein regulation requires new associations to form, but currently we do not know how tolerant the cell is of novel protein interactions. For example, can a nuclear kinase relocate to the cytoplasm without consequence?

Various methodologies have been developed to allow specific affinity-based relocation of proteins in vivo. For example, some systems are designed to disable a location-specific function by sequestering proteins to a specific compartment (*Haruki et al., 2008*; *Robinson et al., 2010*). Alternatively, a leucine zipper-based system was developed to screen for pairwise protein associations, provided that selection for a phenotype is possible (*Devit et al., 2005*). However, none of these approaches have systematically assessed the effects of creating pairwise protein associations, one at a time, across the entire proteome. To address this, we made use of the *Synthetic Physical Interaction* (SPI) system (*Olafsson and Thorpe, 2015*) to create high-affinity interactions between each of the ~six-thousand members of the eukaryotic yeast proteome and target proteins in each of the major cellular compartments. This has allowed us to assay the effect of each of these in vivo binary protein interactions individually upon the normal growth of cells. We find that most protein-protein interactions are benign to the normal growth of cells, but that specific interactions do perturb growth - these interactions are termed Synthetic Physical Interactions or SPIs (*Olafsson and Thorpe, 2015*). The SPIs are enriched for functional regulators, indicating that constitutive colocalization of a regulator with its target causes a growth defect. We are able to use SPIs to identify novel regulatory proteins; for example, we examine SPIs between the kinetochore protein Nuf2 and both Hmo1 and

**eLife digest** Our actions often depend on who we interact with: parents, teachers, friends, colleagues. So it is for proteins in the cell: their function depends on which other proteins they work with. If a protein interacts with new partners or ends up in a new neighborhood of the cell, it can perform an entirely unexpected role, rewiring how that cell works.

There are millions of possible protein-protein interactions, but it is not known how cells behave if their proteins are forced into new associations. For example, how many of these associations affect how well the cell can grow?

Using budding yeast, Berry et al. were able to associate every protein in the cell with proteins from each of the major areas of the cell such as the nucleus, cell membrane or mitochondria. These new associations and relocations were then examined to see how many of them caused problems, slowing the cell's growth or killing it.

Unexpectedly, most forced associations had no detectable effect, indicating that the cell is remarkably tolerant of new protein-protein interactions. This contradicts a common idea that proteins are very fussy about their partner proteins, and will not work properly if they are forced into new interactions.

The associations that do cause a growth defect are often between proteins that normally work together, indicating that their association is normally carefully controlled during the normal growth of cells. In some cases these forced associations identified previously unknown regulators of cell behavior.

Proteins that interact with the wrong partners or are in the wrong place within cells cause a number of diseases. Future forced association experiments will allow us to examine such interactions and possibly search for drugs that will correct the problem.

Sgf29 and find that these two proteins are required to regulate the levels of outer kinetochore proteins. Furthermore, the SPIs correlate with the quaternary structure of large protein complexes such as the kinetochore or nuclear pore. As such, the SPIs provide a powerful tool to complement existing physical and genetic interactions.

## Results

The SPI system uses a GFP-binding protein (GBP) derived from an alpaca antibody (*Rothbauer et al., 2006*), which when fused to a target protein of interest creates binary associations in vivo with GFP-tagged proteins (*Rothbauer et al., 2006*; *Rothbauer et al., 2008*; *Grallert et al., 2013*). We define a target protein as one fused with the GBP and a query protein as one tagged with GFP. By introducing GBP-target proteins into strains encoding GFP-query proteins, we induce an affinity between the target and query proteins via the strong binding of GBP to GFP. We used the *Selective Ploidy Ablation* technique (*Reid et al., 2011*) to introduce a plasmid encoding the GBP-target protein into the collection of ~6000 GFP strains, each of which has a chromosomally integrated GFP introduced at the 3' end of a specific open-reading frame (*Huh et al., 2003*). In each resulting haploid strain, the GBP-target protein is plasmid-encoded and the GFP-query protein is endogenously-encoded; we are therefore able to create a binary protein-protein interaction and assess the effects of this interaction upon growth. We used two independent controls, which were separately transferred into the GFP collection. The first control encodes the GBP alone, and the second encodes the target protein. These two constructs control both for the effects of binding a protein to the GFP tag and also for the ectopic expression of the target gene in each GFP strain. We chose 23 different target proteins that represent 18 of the major cellular compartments (*Figure 1A* and *Figure 1—source data 1*), such as the nucleus (Pus1 and Rad52), the cell membrane (Psr1), and the endoplasmic reticulum (Sec63). The genes encoding these target proteins were fused with GBP and transferred into every strain of the GFP collection (*Figure 1—source data 1*). Thus, for each target protein, we create ~6000 strains each of which contains the target GBP-tagged protein together with a specific GFP-query protein. The effect on growth was assayed by comparing the colony sizes of strains containing the GBP-GFP interaction with the two controls (*Figure 1B,C*)

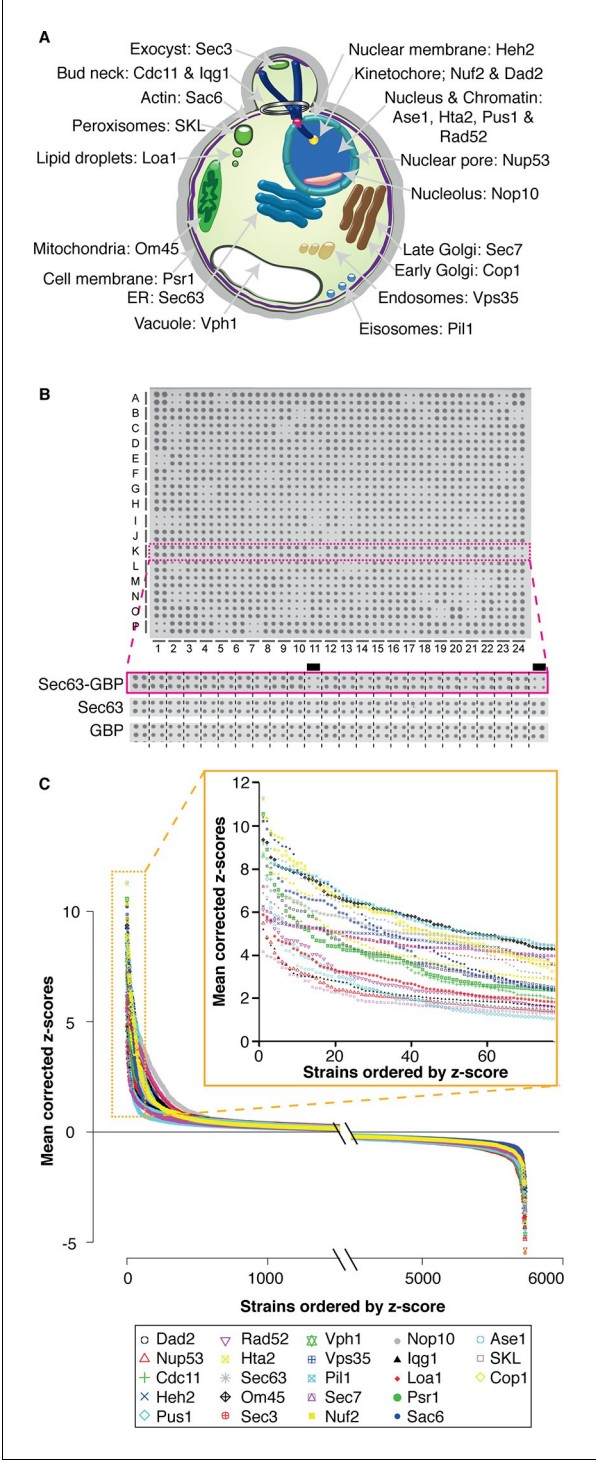

**Figure 1.** Quantitative analysis of the effects of binding proteins throughout the cell. (**A**) A schematic diagram of *S. cerevisiae* indicating the cellular compartments and target proteins within the cell that were associated with each member of the proteome. (**B**) A 1536 colony plate from the Sec63 screen. The inset below shows the highlighted row from the Sec63-GBP plate, the Sec63 control plate and the GBP-only control plates respectively. Growth defects are indicated with a black line. (**C**) The z-scores of all 5734 proteins in each of the 23 screens. For each screen, the strains are ranked according to order of z-scores, positive z-scores indicate a growth defect relative to controls. The inset highlights the strains with the largest growth defects in each screen.

The following source data and figure supplement are available for figure 1:

*Figure 1 continued on next page*

(*Dittmar et al., 2010*). The two controls gave equivalent results (*Figure 1—figure supplement 1* and as previously reported *Olafsson and Thorpe, 2015*) and consequently an average growth score was used.

We expected that many of the forced associations would disrupt cellular homeostasis, but unexpectedly, we found that 98% of GBP–GFP combinations (129,098 out of 131,882) do not affect the growth of cells (*Figure 1C*). These data imply that cells are surprisingly permissive for most protein-protein interactions and as a corollary that cells are broadly tolerant of proteins being relocated within the cell.

In cases where fluorescent imaging was able to detect protein relocalization, we confirmed that ~72% of interactions do occur. Since the GBP tag is linked to red fluorescent protein (RFP), we were able to assay colocalization with GFP. We examined 552 GBP-GFP combinations - each of the 23 GBP-tagged target proteins separately combined with a random selection of 24 GFP-tagged query proteins - using live cell imaging (*Figure 2—source data 1*, for examples see *Figure 2* and *Figure 2—figure supplement 1*). Of the 524 GBP-GFP combinations that we could score, 210 (40%) are already in the same compartment and so we cannot determine whether GFP and GBP associate, of the remaining 314, 225 were detectably colocalized (*Figure 2C*), indicating that in the majority of cases the protein-protein associations do occur (*Figure 2*, *Figure 2—figure supplement 1* and *Figure 2—source data 1*). These observations are therefore consistent with the notion that most synthetic protein-protein interactions do not cause a growth defect.

The microscopy analysis also allows us to examine whether the GBP-tagged target protein recruits the GFP protein to its location or vice versa. We anticipated that each binary protein association would create a 'tug-of-war' between the target protein and the query protein. The image data support this notion; where it is possible to distinguish the location of two proteins in the cell, we observed that there are roughly equal instances of the GBP protein recruiting the GFP protein as the reverse (*Figure 3* and *Figure 3—figure supplement 1*). However, this generalization is not true for some classes/types of proteins. When we look at individual GBP or GFP proteins, we find that structural components more often recruit proteins to their location than enzymes that are not anchored to a specific location (*Figure 3*, *Figure 3—figure supplement 1* and *Figure 2—source data 1*). For example, GFP-tagged cytosolic query proteins such as Cdc55 and Snf1 mostly relocalize to their target proteins (*Figure 2B* and *Figure 3—figure supplement 1*), whereas the GFP-tagged nucleolar proteins Rpa49 and Pwp2 more often recruit GBP-tagged target proteins to their location (*Figure 3—figure supplement 1*). There are some rare cases where the two proteins localize to both locations and also where one or both proteins mislocalize to a new location that is foreign to both (*Figure 2D*). An example of the latter is the recruitment of the nucleosome remodeling protein, Spt6, to the histone subunit Hta2. Constitutive recruitment of a nucleosome remodeler to the chromatin might be expected to give a phenotype and indeed we find that the histone subunit Hta2-GBP is strikingly no longer restricted to the nucleus (*Figure 2E*) concomitant with a strong growth defect. It is possible that we are overestimating the extent of relocalization caused by the GFP-GBP interaction. First, since the target and query proteins are not stoichiometrically matched, some of the GFP or GBP protein will likely remain at its native location. Second, it is possible that in some cases either the GFP tag or the GBP tag is cleaved from its query or target protein respectively, thus giving a false indication of colocalization. It is also possible that imaging underestimates the proportion of relocalization, since we could not score the 210 combinations where proteins are already in the same compartment, these are perhaps more likely to associate via the GFP-GBP interaction. Furthermore, it should be noted that in some cases where we could not detect that the GFP and GBP proteins were colocalized, there was nevertheless either a growth phenotype or a change in the location of one of the proteins. For example, of the 15 Iqg1 associations that failed to show protein colocalization (*Figure 3*), 14 show mislocalization of either the Iqg1 target protein or the GFP query protein.

Around 2% of the forced interactions restrict growth (*Figure 1C*, *4A* and *Figure 1—source data 1*). However, we note that of the 6000 GFP-tagged proteins used in this study, only ~4000 have been validated and are clearly observable (*Huh et al., 2003*). We therefore reanalyzed the proteome-wide data using only 3905 GFP strains with unambiguous fluorescence signal (*Tkach et al., 2012*) and find that ~3% restrict growth (*Figure 4—source data 1*), consistent with the notion that most protein-protein associations do not restrict growth. We did not use a specific threshold cutoff to define a SPI, rather we confirmed the SPIs with the greatest impact on cell growth for each GBP

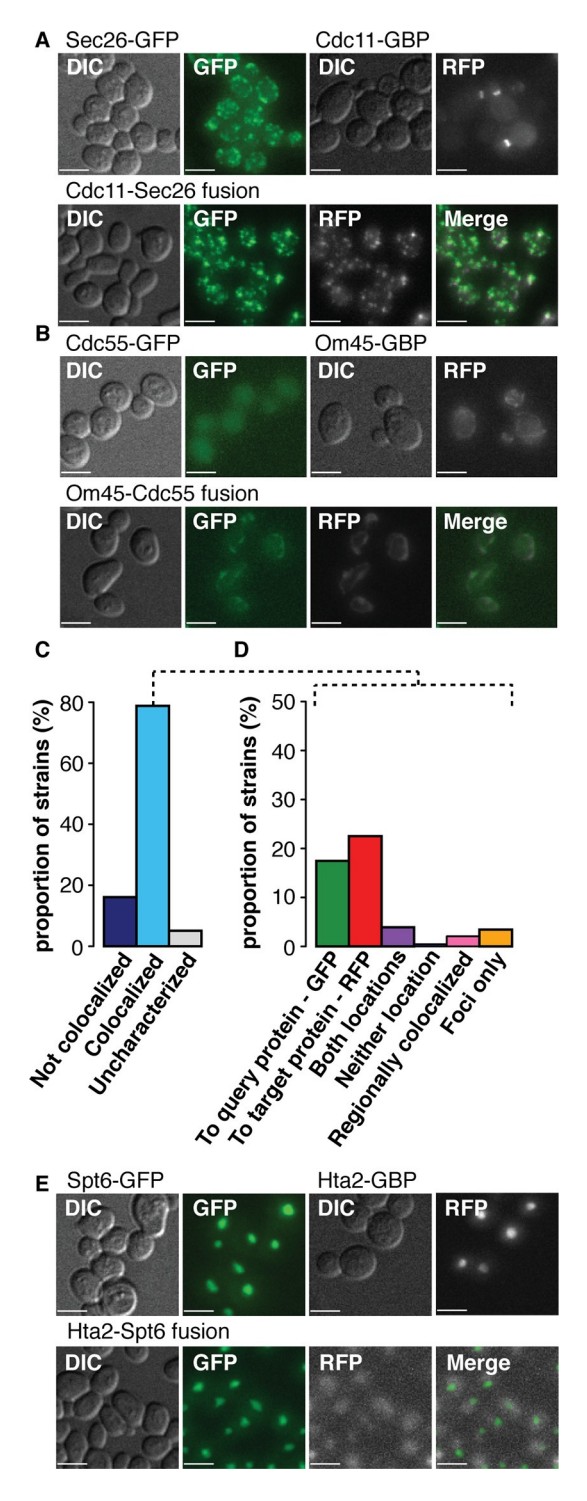

**Figure 2.** Colocalization of target GBP protein and query GFP proteins. (**A**) Cdc11-GBP relocalizes to the Golgi when bound to Sec26-GFP. (**B**) Cdc55-GFP relocalizes to the mitochondria when bound to Om45-GBP. (**C**) Bar chart of the proportion of colocalization (n=552), note that the colocalized category includes 210 combinations where the target and query proteins are within the same compartment and so protein-protein association will not be apparent from this microscopy analysis. (**D**) Bar chart of the direction of movement of GFP and GBP (n=225). 'To query protein - GFP' indicates relocation of the majority of the GBP target protein to GFP (see **A**); 'To target protein - RFP' denotes relocation of the majority of the GFP query proteins to the GBP-RFP target (see **B**). 'Both locations' indicates that GBP and GFP proteins are in both their normal location and those of the other protein
*Figure 2 continued on next page*

by repeating the assay starting with the strongest interaction and proceeding sequentially through the SPIs until the false discovery rate (FDR) reached 40% (*Figure 4—figure supplement 1*). Associations that produced a growth defect relative to controls with 16 replicates in the confirmation experiments are considered SPIs. Thus, some SPIs result from relatively mild growth defects, as outlined in *Figure 4—source data 1*. We note that the false negative rate may be significant, since we did not test further than the 40% FDR and due to the limitations of measuring growth by colony size. Using this approach, we confirmed 2784 SPIs in total produced by 727 GFP-tagged query proteins with one or more of the 23 target proteins (*Figure 4—figure supplement 2* and *Figure 4—source data 1*).

One possible cause of the SPIs is that the target protein would sequester the GFP-tagged query protein away from its normal location. Should this be the case, we would expect low-abundance proteins to be more susceptible to growth defects. However, this is not generally the case for most SPIs, consistent with our earlier findings (*Olafsson and Thorpe, 2015*), since we found there was no correlation between protein abundance and the z-scores (a relative measure of growth) from the 23 GBP screens ($R^2$ values $\leq 0.004$). To address the issue further we grouped all GFP strains into eight categories based upon the abundance of their GFP proteins, each group has 421 proteins. We then plotted the proportion of GFP strains within each group that produced SPIs with a given GBP target (*Figure 4—figure supplement 3A*). Broadly, there are no abundance categories that are consistently enriched for SPIs with all GBP associations. However, we did note that in some cases the group of most abundant proteins had fewer SPIs than the other groups (for example Hta2 and Sec63, see *Figure 4—figure supplement 3A*). To assess whether the levels of the GBP-tagged protein would influence the SPIs, we altered the GBP-tagged protein levels by virtue of their constitutive copper promoter. The *CUP1* promoter functions in the absence of copper and its expression can be gradually increased by adding copper to the growth media. We confirmed that upon addition of increasing amounts of copper, the levels of the GBP target proteins increased, as assayed by quantitative fluorescence imaging of the RFP tag attached to GBP (*Figure 4—figure supplement 3B*). We then retested 400 GFP strains, representing both high and low abundance proteins, with four different GBP target proteins, two of which had less SPIs with high-abundance proteins than expected (Hta2 and Sec63). The results indicate that increasing the expression of the GBP proteins does not specifically increase the number of SPIs within high abundance categories (*Figure 4—figure supplement 3C*). Nevertheless, we expected that a subset of proteins would be particularly sensitive to the effects of forced association and relocalization and this proved true. When we examine all the 727 SPI query proteins collectively (*Figure 4A*), we find that 75 GFP query proteins produce SPIs with at least 10 of our 23 GBP-tagged target proteins (*Figure 4—figure supplement 2A*). These 'frequent SPI query proteins' are on average of lower abundance than less frequent SPI query proteins (*Figure 4—figure supplement 2B,C*), also they are enriched for essential genes ($\approx 83\%$) and for proteins whose *gene ontology* (GO) terms include RNA metabolism (p-value = $9.26\times10^{-5}$), mRNA polyadenylation (p-value = $1.63\times10^{-9}$), cytoplasmic and nuclear transport (p-values = $1.14\times10^{-8}$ and $1.69\times10^{-7}$, respectively), microtubule nucleation (p-value = $5.09\times10^{-8}$), and spindle pole body (p-value = $3.22\times10^{-8}$). We have previously shown that these interactions are mostly suppressed by having an untagged copy of the query protein present in the cell (*Olafsson and Thorpe, 2015*). In heterozygous diploid strains, the untagged version of the SPI query protein is able to complement for the tagged version of the protein that is mislocalized via its association with the target protein. To confirm that the frequent SPI query proteins fall into this category we retested 41 SPIs from the Nuf2 screen that fall into the frequent SPI query proteins group and 40 from the non-frequent SPI query proteins group. Consistent with our expectation all 41 frequent SPIs are suppressed in heterozygous diploid cells, whereas 15% (6 out of 40) SPIs in the non-frequent group were reproduced in diploid cells (*Figure 4—figure supplement 4*). Thus, we conclude that these frequent SPI query proteins are predominantly those whose essential function is location-dependent and whose sequestration to another compartment results in a growth defect (as is routinely achieved using other systems *Haruki et al., 2008*).

To understand whether associations to similar areas of the cell create growth defects from common sets of query proteins, we compared the SPIs generated from each target protein. Spearman's correlation coefficient analysis (*Lubbock et al., 2013*) indicates that, in specific cases, SPI screens using target proteins from the same cellular compartment give similar SPIs (*Figure 4B*). For example, the Pus1 and Rad52 target proteins, which are both in the nucleus, produce SPIs with a similar set of

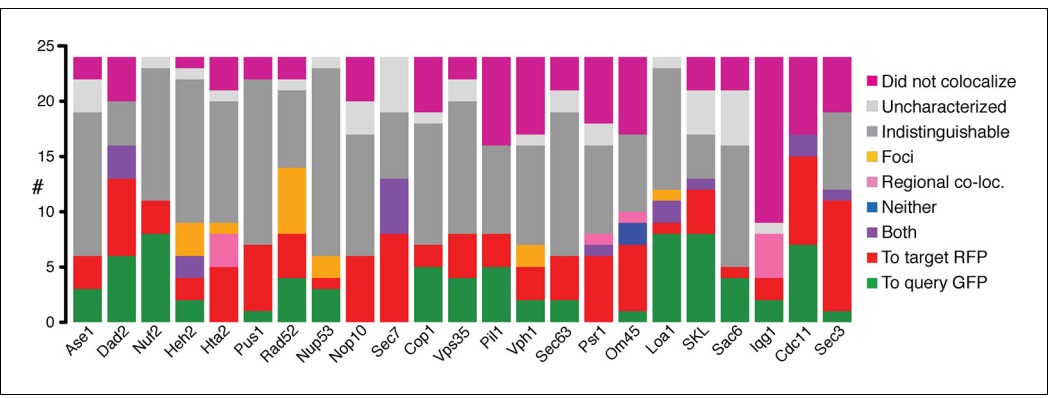

**Figure 3.** Direction of colocalization. (**A**) The proportion of the 24 query proteins that colocalized in the direction indicated. Categories used to characterize the direction of colocalization are described in *Figure 2*. The 'Uncharacterized' category includes strains where there were no cells to image, which is often the case if the interaction perturbs growth.

The following figure supplement is available for figure 3:

**Figure supplement 1.** Direction of colocalization.

query GFP proteins. However, it is interesting to note that some target proteins from the same cellular compartment give quite distinct sets of SPIs. For example, the SPI data for nuclear proteins Nop10 (nucleolus), Heh2 (nuclear membrane), and Hta2 (histone) cluster together but are distinct from both Pus1 and Rad52 (two nuclear enzymes). We suggest that these SPIs segregate into two different classes because Pus1 and Rad52 are non-essential nuclear enzymes, whereas Nop10, Heh2, and Hta2 are structural components, which may be more sensitive to movement. We next asked whether SPI query proteins would be located in the same cellular compartment as their target protein. SPIs between query and target proteins that normally localize to the same cellular compartment are enriched (10.4% of our confirmed SPIs are with target and query proteins from the same compartment, versus an expected value of 7.1% for the full dataset, p-value = $1.8 \times 10^{-9}$, Fisher's Exact test). Also, this notion is true in specific cases, particularly for nuclear proteins. For example, SPIs with a nucleolar protein, Nop10, are enriched for nucleolar components (21 out of 115, p-value = $1.8 \times 10^{-8}$, Fisher's exact test) or SPIs with the microtubule-associated kinetochore component Nuf2, which are enriched for microtubule components (described below). This pattern was typical of nuclear proteins, but not evident for other proteins: for example, the SPIs with the mitochondrial protein Om45 did not include any mitochondrial proteins. However, it should be noted that although there are more SPIs between proteins in the same compartment, SPIs produced by proteins in different compartments tend to give a greater growth defect (*Figure 4C*).

Unexpectedly, we find that SPI query proteins are enriched for characterized physical interactions, compared with non-SPIs (p<2.2 x $10^{-16}$, Wilcoxon's rank-sum). This is visualized by overlaying all the confirmed SPI query proteins onto a graph of the yeast physical interaction dataset (HINT database (*Das and Yu, 2012*), *Figure 4D*). We also asked the same question for each SPI screen using the *Cut-off Linked to Interaction Knowledge* tool (CLIK), which examines quantitative data for interaction density (*Dittmar et al., 2013*). The CLIK tool ranks all genes/proteins by their z-score (high scores bottom left, low scores top right) and then plots the interaction density between all proteins (using data from the Biogrid database *Stark et al., 2006*). If, from a specific target screen, the most growth restricted query proteins are collectively enriched for genetic or physical interactions then a cluster of high density will be visible in the bottom left of the density plot. Most SPI screens have a strong enrichment for genetic and physical interactions indicating that the strongest SPIs share interactions (*Figure 4E* and *Figure 4—figure supplement 5*), which is a predictor of common function. The overlap with physical interactions is particularly surprising; indicating that proteins that normally interact together can induce a growth defect when constitutively bound. Collectively, these observations are

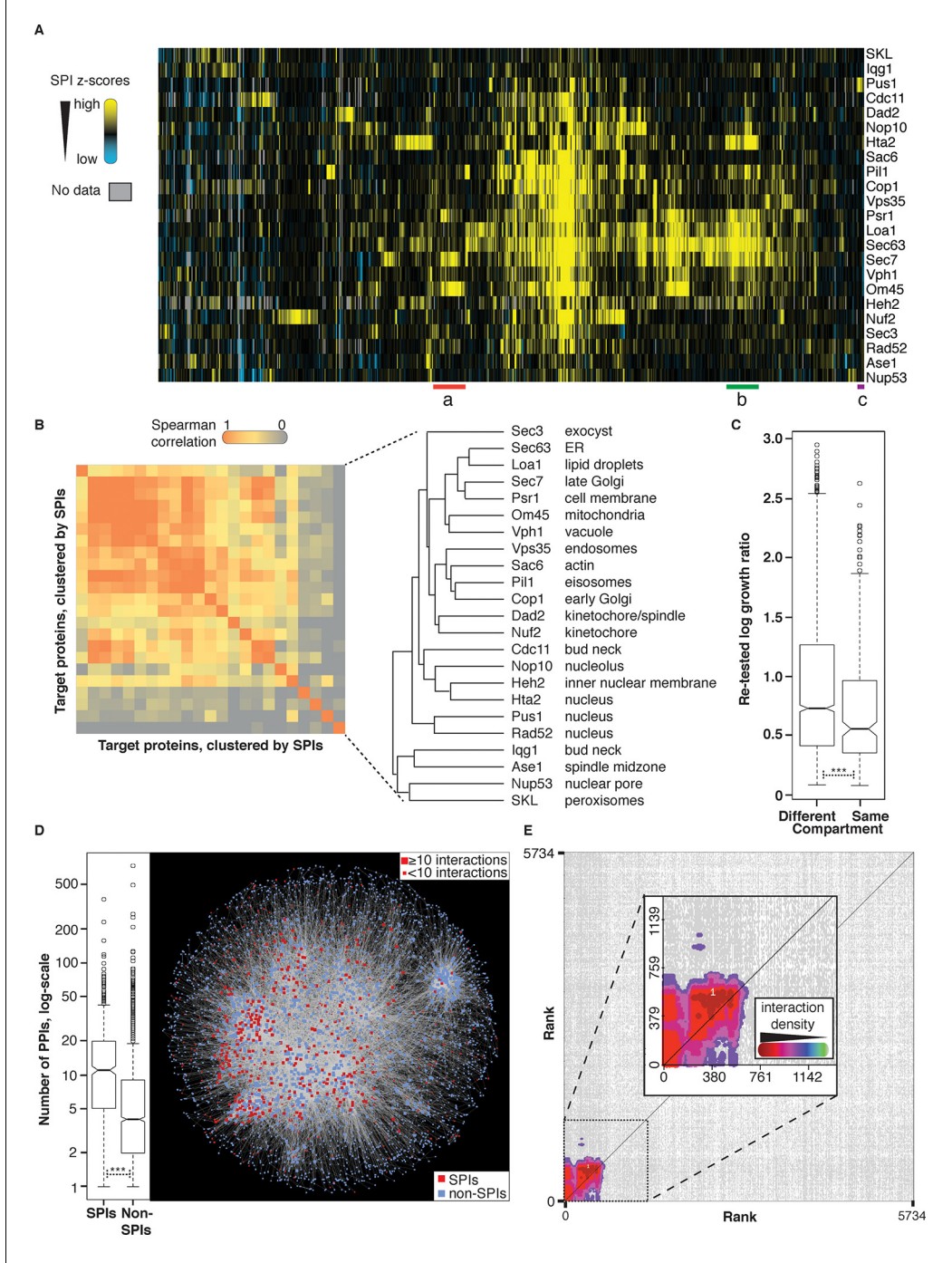

**Figure 4.** Comparisons of synthetic physical interaction screens. (**A**) Cluster analysis of the SPI data. The 23 screens are arranged horizontally and the 727 GFP strains clustered vertically. High z-scores (positive; >2) in yellow and low (negative; < -1) scores in blue. Three distinct clusters are highlighted (a, b, and c) and described in *Figure 4— figure supplement 6*. (**B**) Spearman's Rank Correlation Coefficients for the different SPI screens shows similar compartments give similar SPIs, for example, Sec63 and Loa1 cluster together as do two kinetochore proteins Nuf2 and Dad2. (**C**) The notched box-and-whisker plot indicates the distributions of the retest log growth ratios and indicates that SPIs produced by a query protein and a target protein from different compartments produce stronger growth defects than those from the same compartments (***indicates a p-value = $1.8 \times 10^{-5}$, Wilcoxon's rank-sum). The plot shows the median value (bar) and quartiles (box), the whiskers show the minimum of the range or 1.5 interquartile ranges, outlying data points are indicated as circles and the notches indicate the 95% confidence intervals of the medians. (**D**) The GFP proteins with SPIs have, on average, more protein-protein

*Figure 4 continued on next page*

consistent with the idea that proteins and their regulators are often located within the same compartment, but their temporal or spatial physical association is tightly regulated.

The SPIs for each target protein are also enriched for proteins involved in regulating their function. Gene ontology enrichment analysis for the SPIs demonstrates that specific functional classes of proteins are enriched for each cellular compartment. For example, SPIs for the DNA repair protein Rad52 are enriched for components of the nuclear pore (Ndc1, Nic96, Nup1, Nup85, Nup49, Nup57, Nup84, Nup145, and Nup192; p-value = $7\times10^{-9}$), specifically the Nup84 complex, which functions in specialized types of DNA repair (*Nagai et al., 2008*). Another example is the kinetochore protein Nuf2, whose SPIs are enriched for proteins involved in microtubule organization (Ark35, Bir1, Cbf2, Cdc14, Ctf19, Dad2, Dad4, Dsn1, Ipl1, Kip1, Kip3, Okp1, Spc24, Spc29, Spc42, Spc105, Spc110, Stu1, and Tub4; p-value = $8\times10^{-13}$). Nuf2 is an outer kinetochore protein whose calponin-homology domain directs microtubule binding (*Wei et al., 2007*; *Ciferri et al., 2008*). As such, the SPIs may include numerous novel regulators of their target proteins (*Olafsson and Thorpe, 2015*). To test this, we examined three Nuf2 SPIs in more detail. Hmo1, Sgf29 (both chromatin-associated proteins) and Sst2 (a GTPase activating protein) all gave a strong SPI phenotype with the kinetochore protein Nuf2. Only one of these mutants, *hmo1Δ*, gives a chromosomal instability phenotype (*Stirling et al., 2011*) and none have a reported role in kinetochore function. The SPI data (*Figure 4A*) cluster Hmo1 adjacent to Dad4, an outer kinetochore protein and with other kinetochore proteins (Mcm21, Okp1, Nkp2, Ctf19, and Spc24). To test whether the Hmo1-Nuf2 SPI was unique in the kinetochore, we tested various other kinetochore target proteins fused with GBP in an Hmo1-GFP strain. We find that in addition to Nuf2, Hmo1 has SPIs with Mif2 and Ctf19, but not Kre28, Mtw1, Dad2, Ctf3, Chl4, Skp1, Cnn1, or Cbf1 (*Figure 5A*). These data suggest that the Hmo1 SPI is specific to central/outer kinetochore components. We examined fluorescently tagged kinetochore proteins in *hmo1Δ, sgf29Δ,* and *sst2Δ* cells. We chose two kinetochore proteins, Mtw1 and Dad4, both of which are at the central and outer kinetochore, respectively, and adjacent to Nuf2, Ctf19 and Mif2 and also have been used in quantitative studies (*Joglekar et al., 2006*; *Ledesma-Fernández and Thorpe, 2015*). Strikingly, we find that *hmo1Δ* and, to a lesser extent, *sgf29Δ* cells both have elevated levels of Dad4 outer kinetochore protein associated with their centromeres, although the levels of Mtw1 were unaffected (*Figure 5B,C and 5D*). However, Hmo1 stimulates the activity of the SWI/SNF chromatin remodeling complex (*Hepp et al., 2014*) and therefore may affect expression of the *DAD4* gene. To test whether the *hmo1Δ* mutant was affecting Dad4 protein levels we quantified total cellular Dad4-YFP fluorescence in wild-type and mutant cells and find approximately one third of *hmo1Δ* cells have higher levels of Dad4 than those found in wild-type cells (*Figure 5—figure supplement 1*). Nearly half of the *hmo1Δ* cells have Dad4 levels in the wild-type range (+/- one standard deviation of the wild-type mean); hence cellular Dad4 protein levels are not sufficient to explain the aberrant Dad4 foci seen in most *hmo1Δ* cells (*Figure 5B*). Furthermore, it has previously been shown that Hmo1 is associated with purified kinetochores (*Akiyoshi et al., 2010*), consistent with a direct role at the kinetochore. These data support the notion that in specific cases SPIs define functional regulators.

For each cellular compartment there are relatively few GFP proteins that produce SPIs with just one target protein. The query GFP proteins that produce SPIs have on average 3.8 SPIs with the 23 target proteins. However, those GFP proteins with just one SPI may be informative. For example, the histone subunits Hta2, Htb1, Htb2, and Hhf2 as well as the chromosomal proteins Bub1 and Mft1 have unique SPIs with the eisosome component Pil1. These interactions may indicate a nuclear role for Pil1, which relocalizes from the plasma membrane in response to DNA damage (*Tkach et al., 2012*) and associates with histones and chromosomal proteins (*Lambert et al., 2009*; *Akiyoshi et al., 2010*). Indeed, the Pil1-histone SPIs result from Pil1 recruitment into the nucleus (*Figure 2—figure supplement 2*).

Since selected SPI query proteins are enriched for physical and genetic interactions and contain proteins involved in regulating the biology of their target, we next performed hierarchical clustering analysis in order to test whether SPI data can be used to assess functional associations between proteins (*Figure 4A*). We find that query proteins from specific large functional complexes cluster together, for example, the mediator complex, which is involved in activating transcription, clusters together as do members of the COP1 coatomer, the outer ring of the nuclear pore, the signal recognition particle and TRAMP complex (*Figure 4—figure supplement 6*). It is important to note that SPIs are not a substitute for physical interaction data, but rather represent a common phenotype in

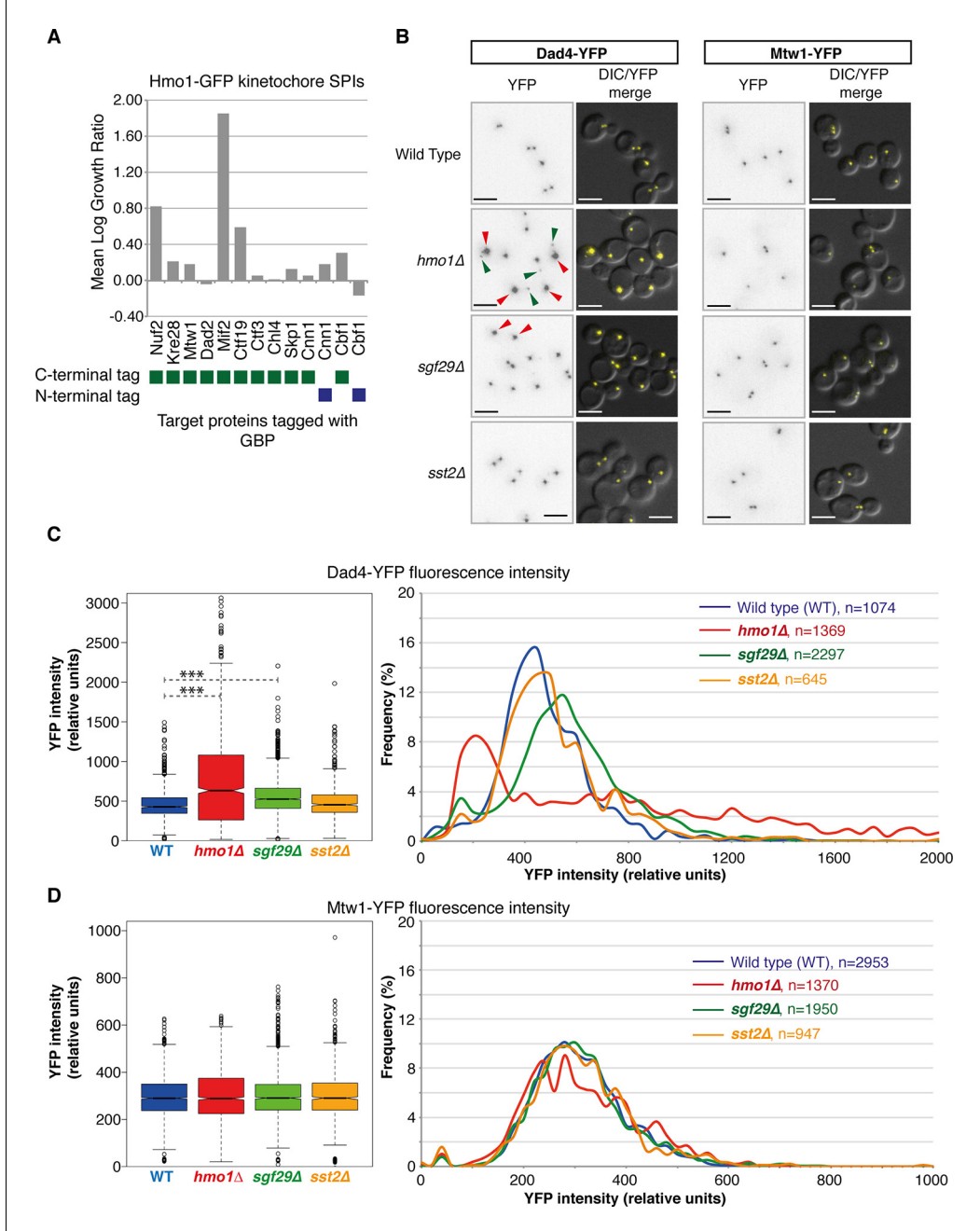

**Figure 5.** Nuf2 SPIs affect kinetochores. (**A**) The Hmo1-GFP query protein encoding strain was transformed separately with 13 plasmids encoding different kinetochore proteins target proteins tagged with GBP (4 replicates each). The growth relative to controls (GBP alone and target protein alone) was assessed as in *Figure 1*. (**B**) Deletion of *HMO1*, *SGF29*, and *SST2* were separately introduced into strains encoding Dad4-YFP and Mtw1-YFP at their endogenous loci. Fluorescence imaging of these strains reveals that *hmo1Δ* mutants have large-bright Dad4-YFP kinetochore foci (red arrows) and some weak foci (green arrows). *sgf29Δ* mutants contain bright Dad4-YFP foci (red arrows). In all cases, there are no effects upon Mtw1-YFP foci (right panels). Scale bars in all images are 5 μm. (**C**) Quantitation of the Dad4-YFP kinetochore foci fluorescence levels from these cells indicates that the levels of Dad4-YFP at kinetochores are affected by deletion of either *HMO1* or *SGF29*. The left notched box and whiskers plot indicates the median (background subtracted) fluorescence values of kinetochore foci in relative units. The plot shows the median value (bar) and quartiles (box), the whiskers show the minimum of the range or 1.5 interquartile ranges, outlying data points are indicated as circles (note that several outlying data points are not shown as they are beyond the scale of the plot). The notches indicate the 95% confidence intervals of the medians (***indicates p-values <10$^{-10}$ from a Wilcoxen's rank-sum test). It should be noted that the distribution of

*Figure 5 continued on next page*

Berry *et al*. eLife 2016;5:e13053. DOI: 10.7554/eLife.13053

response to forced association. Collectively, the clustering of protein complexes, gene ontology enrichment and physical and genetic enrichment indicate that specific target proteins show SPIs with sets of query proteins that share a common location, potentially common components of larger protein complexes. Thus, although the proteome-wide SPI data themselves do not directly give structural information, the SPI data groups query proteins within these known protein complexes.

We next asked whether the SPI data would correlate with the quaternary structure of multi-protein complexes, since protein associations with one part of a complex may give a similar growth phenotype that contrasts with a different part of that same complex. We chose the kinetochore as an example, since this is a large array of between 60 and 100 proteins that are arranged into defined sub-complexes (*Biggins, 2013*). We selected these proteins (and some kinetochore-associated proteins) and clustered them based upon their SPI scores from the 23 screens. We find that key sub-complexes within the kinetochore are clustered together purely based upon their 23 SPI scores (*Figure 6*). For example, three of the four members of the COMA complex cluster together (Ctf19, Okp1, and Mcm21) with two members of the Ctf3 complex (Mcm22 and Nkp2), and Cse4 and Chl4, which are all part of the constitutive centromere associated network (CCAN) of inner kinetochore proteins that bind to centromeric DNA. Three of the four MIND complex members (Dsn1, Nnf1, and Nsl1) also cluster with Spc24, Kre28 and Nuf2, which are all part of the KMN network of outer kinetochore proteins. In contrast, the DAM/DASH complex, which is composed of 10 different proteins, segregates into distinct clusters (with Dad2, 3, and 4 distinct from Dam1, Ask1, Dad1, Spc34, and Duo1). Dad2, 3 and 4 are small central domain subunits of the DAM/DASH complex that are important for structural integrity of the complex and therefore potentially sensitive to association with other proteins (consequently they have many SPIs). In contrast Dam1, Duo1, and Spc34 are key interaction hubs for the decameric complex (*Shang et al., 2003*) and Ask1's C-terminus plays an important role in intercomplex interactions (*Ramey et al., 2011*). Thus these proteins form external surfaces on the complex, which may be more tolerant of protein association. A similar correlation with the quaternary structure can be made for another large protein assembly, the nuclear pore complex (*Figure 6—figure supplement 1*). Hence, although SPIs do not substitute for physical interaction data they indicate a common phenotype produced by specific protein-protein associations.

## Discussion

The SPI technology has allowed us to create binary protein associations throughout the cell and in many cases these interactions result in protein relocalization. However, only a small fraction of these interactions lead to a measurable growth phenotype, suggesting that cells are highly tolerant of both protein mislocalization and protein-protein associations. There are exceptions, proteins that do affect growth in almost any location. For example the ubiquitin hydrolase, Doa4 and numerous proteins involved in transport (*Figure 4—source data 1*). Furthermore, there are proteins whose association with specific proteins causes a growth defect. We find that these SPIs are enriched for proteins that physically interact (*Figure 4*). Collectively the SPI data allow us to both identify regulatory proteins (*Olafsson and Thorpe, 2015* and *Figure 5*) and provide information on quaternary structure of specific large complexes within the cell (*Figure 6*). These data illustrate that SPIs can be used, like physical interactions, to reveal the functional organization of the cell. However, since the readout of SPIs is phenotypic, in this case cell growth, the SPIs indicate functional interactions rather than physical interactions per se. Thus, the SPI methodology provides a powerful in vivo proteomics tool to map the mechanisms underlying spatial regulation within cells. The SPI technology may be particularly informative to define interactions that are detrimental under conditions of stress, drug treatment or other specific cellular perturbations. Many disease pathologies result, at least in part, from the mislocalization of proteins in cells (*Hung and Link, 2011*). Recent studies are discovering the extent to which specific drugs induce global changes in protein location (*Tkach et al., 2012*; *Breker et al., 2013*; *Chong et al., 2015*). Combining this cellular pharmacodynamics knowledge with SPI data opens the possibility of using drugs to induce therapeutic changes in protein localization; of the 727 SPI query proteins identified here, ~76% (549) have human homologs compared to 56% (3766) of the whole yeast genome (6604 ORFs) (YeastMine, *Balakrishnan et al., 2012*). This study provides the first comprehensive map of the effects of forced protein associations within cells.

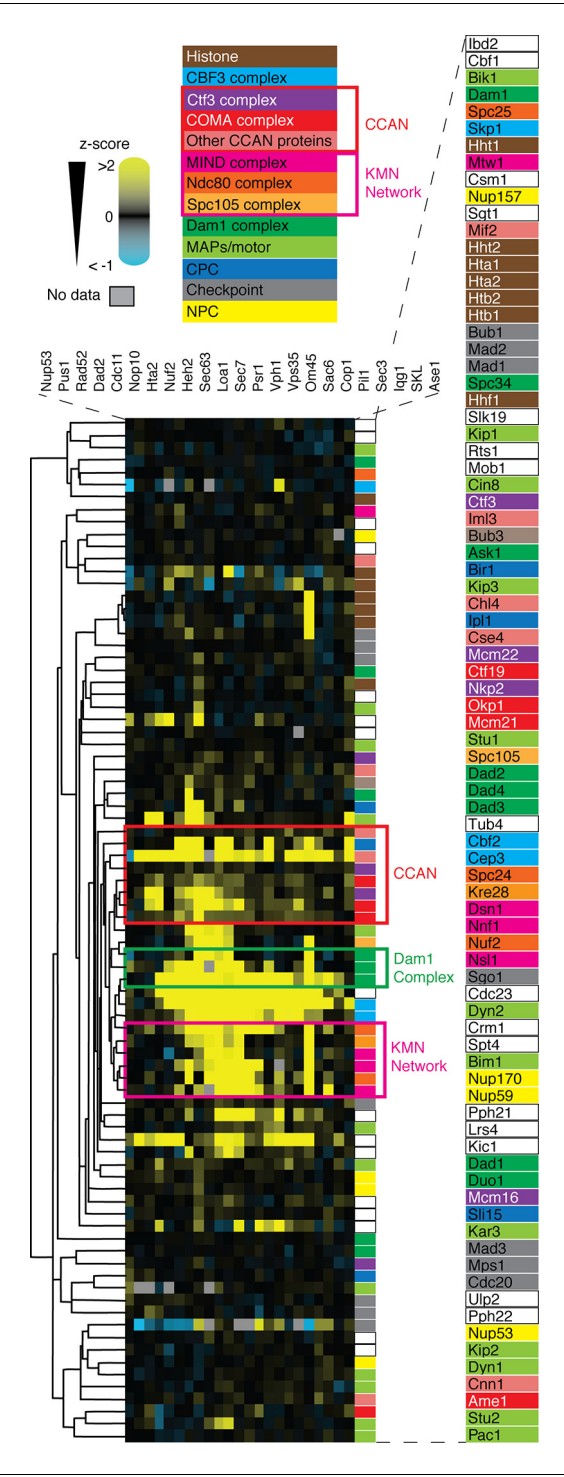

**Figure 6.** Cluster analysis of kinetochore and associated proteins using the SPI data are plotted as a heat-map. High z-scores (positive; >2) are shown in yellow and low (negative; < -1) scores in blue (as in *Figure 4A*). The different protein complexes within the kinetochore are color-coded as indicated in the legend. Based on the SPI data alone, key complexes within the kinetochore cluster together as indicated by the colored boxed regions of the plot.

The following figure supplement is available for figure 6:

**Figure supplement 1.** Clustering analysis of nuclear pore complex (NPC).

## Materials and methods

### Yeast strains and methods

All yeast strains used in this study are listed in *Table 1*. W303 strains are $ADE2^+ RAD5^+$ derivatives of W303 (*can1-100 his3-11,15 leu2-3,112 ura3-1* unless otherwise indicated *Thomas and Rothstein, 1989*; *Zou and Rothstein, 1997*). GFP strains are all based upon BY4741 (*his3Δ1 leu2Δ0 met15Δ0 ura3Δ0 Brachmann et al., 1998*; *Huh et al., 2003*). Yeast were grown in standard growth medium including 2% (weight/volume) of the indicated carbon source (*Sherman, 2002*). Yeast plasmids were created using the gap-repair cloning technique, which combines a linearized plasmid with PCR products using in vivo recombination. All PCR products were generated using primers from Sigma Life Science and PfuII Ultra proof reading polymerase (Agilent Technologies, UK) or Q5 polymerase (New England Biolabs, USA). All plasmid constructs (listed in *Figure 1—source data 2*) were validated using Sanger sequencing (Beckman Coulter Genomics, UK).

### Selective ploidy ablation (SPA) screening

The SPA screening method is a mating-based approach for yeast transformation, and we followed the established protocol (*Reid et al., 2011*). The SPA method relies upon a universal donor strain (UDS, W8164-2B) that includes conditional centromeres on each and every chromosome. This strain is transformed with a plasmid encoding the GBP-tagged target protein (or controls) and then mated en masse with the collection of GFP strains. The resulting diploids are converted back to haploids by first destabilizing and then counter-selecting against all of the chromosomes from the UDS. The resulting colonies are then assessed for growth by measuring colony size as described below. In the first step, plasmid constructs (encoding GBP alone, target protein alone or target-GBP) were transferred into the UDS by transformation. The three resulting strains were separately mated with arrays of *MAT*a GFP strains (*Huh et al., 2003*) on YPD agar plates for 24 hr. The resulting colonies were then copied to synthetic galactose medium lacking leucine to destabilize the donor chromosomes for 24 hr. Finally, colonies were copied onto galactose medium lacking leucine, including the drug 5-Fluoroorotic acid (5-FOA) to counter-select against the UDS chromosomes. Plates were then grown at 30°C for 48–72 hr prior to imaging. All mating and copying of yeast colonies utilized a RoToR pinning robot (Singer Instruments, UK) with a minimum of four replicates per strain.

### Quantitative analysis of high-throughput yeast growth

After SPA screening, the resulting agar plates were scanned using a desktop flatbed scanner (Epson V750 Pro, Seiko Epson Corporation, Japan) at 300 dpi resolution in transmission mode. These images were processed and analyzed using the ScreenMill suite of software (*Dittmar et al., 2010*), which assesses growth based upon the two-dimensional size of the colonies. The software was run in default mode, both for the kinetochore-specific screen and for the proteome-wide screen. For retesting strains for growth defects, plate images were normalized using specific controls on the plate as a reference, rather than the default plate median. This is necessary when the majority of the strains on a plate are affected since this will influence the plate median.

### Spatial smoothing algorithm

Colonies arrayed on agar plates often grow faster on one side of the plate than the other. This growth effect can be caused by temperature or humidity gradients within incubators, variable thickness of agar (and hence concentration of nutrients), or uneven pinning pressure during plate copying. These anomalies can result in one side of the plate producing an overall higher z-score than the other. To correct for these type of biases, algorithms adjust colony size data to reflect overall even growth across a plate (*Collins et al., 2006*; *Baryshnikova et al., 2010*). The ScreenMill suite of software used for our analysis does not contain such corrections and so we employed a simple algorithm to correct z-scores for spatial anomalies (*Olafsson and Thorpe, 2015*).

### Fluorescence microscopy

To examine the levels and location of tagged proteins within the cells, we used epifluorescence microscopy. Log phase cells were embedded in 0.7% low melting point agarose dissolved in the appropriate growth medium. The depth of agarose between the slide and coverslip is fixed at 6–

**Table 1.** Yeast strains used in this study.

| Strain name | Genetic background | Relevant genotype | Reference |
|---|---|---|---|
| W8164-2B | W303 | MATα CEN1-16::Gal-Kl-URA3 | (Zou and Rothstein, 1997) |
| GFP strains | BY4741 | MATa his3Δ1 leu2Δ0 met15Δ0 ura3Δ0 XXX-GFP::HIS3 | (Huh et al., 2003) |
| PT147-7C | W303 | MATa TRP1 lys2Δ DAD4-YFP::NAT SPC42-RFP:: | This study |
| PT12-13D | W303 | MATa TRP1 MTW1-YFP hmo1Δ::KAN | This study |
| T403 | W303 | MATa TRP1 lys2Δ DAD4-YFP::NAT SPC42-RFP::HYG hmo1Δ::KAN | This study |
| T404 | W303 | MATa TRP1 lys2Δ DAD4-YFP::NAT SPC42-RFP::HYG sgf29Δ::KAN | This study |
| T402 | W303 | MATa TRP1 lys2Δ DAD4-YFP::NAT SPC42-RFP::HYG sst2Δ::KAN | This study |
| T406 | W303 | MATa TRP1 MTW1-YFP hmo1Δ::KAN | This study |
| T407 | W303 | MATa TRP1 MTW1-YFP sgf29Δ::KAN | This study |
| T405 | W303 | MATa TRP1 MTW1-YFP sst2Δ::KAN | This study |

8μm, slightly larger than the diameter of the average yeast cell, which maintains a consistent distance from the coverslip to the cell nucleus. Cells were imaged with a Zeiss Axioimager Z2 microscope (Carl Zeiss AG, Germany), using a 63x 1.4NA oil immersion lens, illuminated using a Zeiss Colibri LED illumination system (GFP=470 nm, YFP=505 nm, and RFP=590 nm). Bright field contrast was enhanced with differential interference contrast (DIC) prisms. The resulting light was captured using either a Hamamatsu ORCA ERII CCD camera containing an ER-150 interline CCD sensor with 6.45 μm pixels, binned 2x2 (Hamamatsu Photonics, Japan) or a Hamamatsu Flash 4 Lte. CMOS camera containing a FL-400 sensor with 6.5 μm pixels, binned 2x2. The exposure times were set to ensure that pixels were not saturated and were identical between control and experimental images. All images were acquired using either Axiovision or Zen software from Zeiss. Images shown in the figures were prepared using Volocity imaging software (Perkin Elmer Inc., USA) and control and experimental images have identical linear contrast adjustments unless otherwise stated.

## Fluorescence image analysis

To quantify the relative amount of RFP in cells containing GBP-RFP tags we used custom scripts for the Volocity image analysis software (Perkin Elmer Inc. USA). Briefly, red fluorescence regions were identified within the three-dimensional images based upon an intensity threshold. These regions were then dilated by a fixed amount (~600 nm) in each direction to ensure that we assay all of the red fluorescence signal. The regions were further dilated (2.4 μm) to create an outer background region, which was subtracted from each fluorescence measurement (the script is available online https://sourceforge.net/projects/berry-et-al/files/RFP_quantitation.assf/download).

To quantify the relative levels of Dad4-YFP and Mtw1-YFP kinetochore proteins within kinetochore foci, we employed a custom ImageJ script (Ledesma-Fernández and Thorpe, 2015). To quantify the total cellular levels of Dad4-YFP we measured the YFP fluorescence signal from maximum projection images (from a stack of vertically separated z planes) for each cell and subtracted a mean background signal specific to each image (this script is available at https://sourceforge.net/projects/berry-et-al/files/general_cell_quan.ijm/download).

## Bioinformatics analysis

Michael Eisen's cluster program (version 3.0) was used to cluster the SPI data (Eisen et al., 1998). We used hierarchical centroid linkage clustering of both the GBP screens and the GFP-tagged genes. For the quaternary structure examples (Figure 6, Figure 4—figure supplement 6 and Figure 6—figure supplement 1) only a selected subset of the GFP strains were used for the cluster analysis. Cluster diagrams were visualized using Java Treeview (Saldanha, 2004). Gene ontology enrichment analysis was performed using the GOrilla algorithm (cbl-gorilla.cs.technicon.ac.il [Eden et al., 2009]).

## Acknowledgements

This work was funded by a Medical Research Council U.K. Centenary Award and research grant (MC_UP_A252_1027). The Francis Crick Institute is funded by The Medical Research Council UK, Cancer Research UK, the Wellcome Trust, Imperial College London, University College London and Kings College London. We thank D Peer, H Caulston, G Brown, B Andrews, E Styles, R Rothstein, J Dittmar, R Reid, I Overton, J Bahler, D Gresham and M Gartenburg. The authors declare no competing financial interests.

## Additional information

### Funding

| Funder | Grant reference number | Author |
| --- | --- | --- |
| Medical Research Council | MC_UP_A252_1027 | Peter H Thorpe |
| Cancer Research UK | Institute core funding | Peter H Thorpe |
| Wellcome Trust | Institute core funding | Peter H Thorpe |

The funders had no role in study design, data collection and interpretation, or the decision to submit the work for publication.

### Author contributions

LKB, GÓ, PHT, Conception and design, Acquisition of data, Analysis and interpretation of data, Drafting or revising the article; EL-F, Acquisition of data, Analysis and interpretation of data, Drafting or revising the article

### Author ORCIDs

Peter H Thorpe, http://orcid.org/0000-0003-1649-6816

## Additional files

### Major datasets

The following previously published datasets were used:

| Author(s) | Year | Dataset title | Dataset URL | Database, license, and accessibility information |
| --- | --- | --- | --- | --- |
| Das J, Yu H | 2012 | High-quality protein interactomes and their applications in understanding human disease. | http://hint.yulab.org/data/SaccharomycesCerevisiaeS288C_binary_hq.txt | Publicly available at the High-quality Interactomes website (http://hint.yulab.org/) |
| Stark C, Breitkreutz BJ, Reguly T, Boucher L, Breitkreutz A, Tyers M | 2006 | BioGRID: a general repository for interaction datasets. | http://thebiogrid.org/download.php | Publicly available at the BioGRID website (http://thebiogrid.org/) |

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
