## [Decision Letter]

Thank you for submitting your work entitled "Synthetic protein interactions reveal a functional map of the cell" for consideration by *eLife*. Your article has been reviewed by three peer reviewers, and the evaluation has been overseen by Randy Schekman as the Senior and Reviewing Editor.

The following individuals involved in review of your submission have agreed to reveal their identity: Stanley Fields and Nevan Krogan (peer reviewers).

The reviewers have discussed the reviews with one another and the Reviewing Editor has drafted this decision to help you prepare a revised submission.

Summary:

The authors present a screen using their newly developed SPI system, which allows for the creation of artificial interactions between pairs of proteins. They create binary interactions between each of the ~6,000 yeast proteins and 23 target proteins that represent the major cellular compartments. The effects of these interactions on growth are assayed by measuring colony sizes. In spite of these interactions often leading to protein relocalization, they find only a small fraction lead to a measurable growth phenotype, suggesting that the cells are tolerant of both protein movement and association. The authors highlight how their method can be used to discover new regulatory relationships and to provide structural information on large cellular complexes.

The SPI system provides an exciting complement to PPIs and genetic interactions, and the scale of the collected dataset is impressive. The manuscript is well written and for the most part the methods are clearly described. Together, the importance and quality of the work makes it suited for publication in *eLife* as a tool/resource. However, there are a number of important points that need to be addressed.

Essential revisions:

1) The analysis on a limited number (24) of GFP-fusion proteins suggests that for only roughly ~20% of time when they co-express a GFP-fusion query protein with a target protein do they see mislocalization of the GFP fusion outside of where it is normally found. But even this may be a large over estimation of the degree to which their system is causing protein mislocalization. First, they do not evaluate what fraction of the GFP-fusion is mislocalized. Second, cleavage of the GFP from the fusion protein will result in mislocalization of the GFP domain but not of the rest of the protein. Finally, they report on ~6000 individual GFP fusion proteins obtained from the library developed in Huh et al. (2003). But in that paper only ~4500 of the fusion proteins were validated and observable. Thus unless the remaining 1500 strains were made and validated separately, they are suspect. In the end because of these and other concerns, I think it is not possible evaluate from these data what fraction of the proteins tolerate being effectively mislocalized to a cellular location that is different from where it is naturally found. This limitation must be discussed and explained.

2) The potentially most interesting result reported here is the detection of Synthetic Physical Interactions (SPIs), where a forced interaction does cause a growth phenotype. But it's not entirely clear that one learns all that much from these results. The one example of an SPI that they do examine in more detail in Figure 5 could easily be an indirect relationship rather than a direct molecular link (yeast null for chromatin associated proteins, *hmo1* and *Sgf29*, show increase levels of a kinetochore protein Dad4). This would be much more compelling if the authors had data indicating some direct physical interaction (IP/mass spec) or biochemical rational for how *hmo1* could regulate the kinetochore to support their claim that SPIs can identify functional regulators- this could easily be a case where *hmo1* regulates transcription of Dad4 directly or indirectly. If so, please include this in the revised version or at least comment on relevant data in support of a direct and/or functional interaction.

3) The clustering of complexes (Figure 6) is not that convincing. the approach seems like an awkward strategy for obtaining information that is much more effectively obtained by mass-spec. For example, the histones fall into two distinct categories and the CPC, NDC80, Spc105 and MAPs complexes don't seem to cluster at all. This seeming disparity with known interactions requires explanation.

4) Results, second paragraph: “we found that 98% of GBP-GFP combinations […] do not affect the growth of cells.” Do 98% not affect growth at all, or are they using a z-score threshold? If z-score, this should be clarified, as it is not necessarily the same as "no effect on growth".

5) Results, third paragraph and Figure 2: “Fluorescent imaging confirmed that ~83% of interactions do occur and typically result in protein relocation […] Of the 524 GBP-GFP combinations that we could score, 435 were detectably colocalized (Figure 2), indicating that in most cases the protein-protein fusions do occur.” Figure 2 shows this and 2D breaks it down in more detail. However, ~50% of the colocalized strains belong to the "indistinguishable" category = proteins that normally colocalize (2D). It is misleading to include this category for estimating how often relocation/interaction occurs. Correcting for this will result in a percentage dramatically lower than 83%. Similarly, this should be clarified in the counts in the text ("435 of 524") and the Figure 2 legend.

6) Results, fifth paragraph: “Around 2% of the forced interactions restrict growth […]” Specify z-score cutoff used to qualify as restriction in main text.

7) Results, fifth paragraph: “[…]of the 727 SPI proteins […]” -> of the 727 SPI query proteins.

8) Results, fifth paragraph: “[…] whose sequestration to another compartment is lethal […]” Did they show it's actually lethal or just decreases growth (i.e. sick)?

9) Results, fifth paragraph: Discuss why there is a difference in suppression of interactions between the frequent and non-frequent SPI groups.

10) Results, sixth paragraph and Figure 4: It's here stated that target proteins from the same cellular compartment give similar SPIs. While the figure and analysis are suggestive of this, it would be better to carry out a statistical test to corroborate the statement. E.g. Box plots of the distributions of SPIs from same vs. different compartments, accompanied by a Wilcoxon rank-sum test to determine significance.

11) Results, seventh paragraph and Figure 4: The authors are comparing two distributions of PPI counts (SPI vs. non-SPI) and compute a p-value for the difference using Spearman's rank correlation. I doubt Spearman's rank correlation can be used to produce a p-value for the difference between two distributions. Additionally, the stated p-value (2.2E-16) appears extremely optimistic given the large overlap of the box plots.

12) Results, seventh paragraph and Figure 4: The CLIK analysis should be described better.

13) Results, eighth paragraph: Describe or reference the gene ontology enrichment analysis.

14) Results, eighth paragraph: Was fluorescently tagged Nuf2 examined in these cells as well? How is Nuf2 affected?

15) Figure 5: Explain why Mtw1 was chosen.

16) Figure 5: Describe in much more detail how to interpret these plots. Also, what are the error bars? Finally, is the p-value (E-10) really correct? Depending on what the error bars represent this looks low.

17) Results, last paragraph: “[…] SPI data can be used to predict protein complexes […]” Substantiate this claim. Show if the SPI data actually have predictive power with a ROC or precision recall curve.

18) Results, last paragraph: Try to interpret why the DAM/DASH complex segregates into distinct clusters.

19) Discussion: I suggest explicitly writing "physical interactions" instead of "interactions" to not confuse with SPIs.

20) Discussion: “[…] and derive quaternary structure […]” ‘provide information on’ would be more appropriate than ‘derive’.

21) Discussion: “[…] 54% (394) are conserved in human cells.” May be worth discussing how this compare to the conservation of the complete yeast genome to human?

[Editors' note: further revisions were requested prior to acceptance, as described below.]

Thank you for submitting your article "Synthetic protein interactions reveal a functional map of the cell" for consideration by *eLife*. Your article has been reviewed by three peer reviewers, and the evaluation has been overseen by Randy Schekman as the Senior and Reviewing Editor. Stanley Fields and Nevan Krogan have agreed to share their identity. There remains one concern with the language in your Abstract. Please adjust this according to the suggestion of reviewer #1.

*Reviewer #1:*

The authors have largely addressed the concerns I raised and put in appropriate caveats in their revised manuscript. I am still concerned that their Abstract is misleading in claiming to establish that proteins have an "unanticipated tolerance for forced protein associations and consequently their relocation". I think it is likely that in many, perhaps the large majority, of the cases where apparent relocation of a protein does not disrupt function has to do with at least partial retention of the protein in its correct locations. It would indeed be quite surprising if most nuclear proteins could function in the cytosol or most organellar localized proteins could function outside of their native organelle. I do not think this is what the authors intend to say (and certainly is not what they have shown) but I could easily see how a casual reader of the Abstract could be left with this impression.

*Reviewer #2:*

I find the revised version acceptable.

*Reviewer #3:*

I am happy with the revisions and support publication.

---

## [Author Response]

*Essential revisions: 1) The analysis on a limited number (24) of GFP-fusion proteins suggests that for only roughly ~20% of time when they co-express a GFP-fusion query protein with a target protein do they see mislocalization of the GFP fusion outside of where it is normally found. But even this may be a large over estimation of the degree to which their system is causing protein mislocalization. First, they do not evaluate what fraction of the GFP-fusion is mislocalized. Second, cleavage of the GFP from the fusion protein will result in mislocalization of the GFP domain but not of the rest of the protein. Finally, they report on ~6000 individual GFP fusion proteins obtained from the library developed in Huh et al. (2003). But in that paper only ~4500 of the fusion proteins were validated and observable. Thus unless the remaining 1500 strains were made and validated separately, they are suspect. In the end because of these and other concerns, I think it is not possible evaluate from these data what fraction of the proteins tolerate being effectively mislocalized to a cellular location that is different from where it is naturally found. This limitation must be discussed and explained.*

We agree that it is possible that the imaging overestimates the frequency of mislocalization. We have now discussed these limitations (Results, fourth paragraph). The amount of a given protein that is mislocalized will depend in part on stoichiometry of the GBP and GFP tagged proteins, which is why we examined the effects of stoichiometry in Figure 4—figure supplement 2 and Figure 4—figure supplement 3. We have made clear that not all the protein is likely to be mislocalized. However, where we score the protein to be mislocalized, this is because the majority of the fluorescence signal has changed location, we have made this clear in the legend of Figure 2. The 24 GFP proteins and 23 GBP tagged proteins tested by imaging show clear localization and have not been reported to be cleaved, however, we cannot rule out cleavage, particularly when the GFP- and GBP-tagged proteins are associated. We have now made clear the cleavage of the GFP protein from the C-terminus of a protein (or cleavage of the GBP from the target protein) will affect these data (in the aforementioned paragraph). Additionally, we have reanalyzed the global data, excluding nearly 2000 GFP strains where protein levels are low or the signal has not been validated. We find that the proportion of associations that affect growth are 3%, which is similar to the data for the full set of GFP-tagged protein, this analysis is now discussed (Results, fifth paragraph), top section, with details included in [Supplementary-material SD4-data].

*2) The potentially most interesting results reported here is the detection of Synthetic Physical Interactions (SPIs), where a forced interaction does cause a growth phenotype. But it's not entirely clear that one learns all that much from these results. The one example of an SPI that they do examine in more detail in Figure 5 could easily be an indirect relationship rather than a direct molecular link (yeast null for chromatin associated proteins, hmo1 and Sgf29, show increase levels of a kinetochore protein Dad4). This would be much more compelling if the authors had data indicating some direct physical interaction (IP/mass spec) or biochemical rational for how hmo1 could regulate the kinetochore to support their claim that SPIs can identify functional regulators- this could easily be a case where hmo1 regulates transcription of Dad4 directly or indirectly. If so, please include this in the revised version or at least comment on relevant data in support of a direct and/or functional interaction.*

To characterize the effect of *hmo1∆* mutants in more detail, we examined whether the Hmo1-Nuf2 SPI is specific to Nuf2 or more general to the kinetochore, we have now tested 10 other kinetochore GBP-target proteins and find that in addition to Nuf2, both Mif2 and Ctf19 give a SPI, whereas Kre28, Mtw1, Dad2, Ctf3, Chl4, Skp1, Cnn1 and Cbf1 do not (new Figure 5) – suggesting that the effect of Hmo1 is restricted to specific regions within the central/outer kinetochore. We have now indicated that Hmo1 was identified by mass spectroscopy from immuno-precipitation of kinetochore complexes (work from Sue Biggins’ lab, Results, eighth paragraph). However, it remains possible that increased *DAD4* expression is contributing to the phenotype. In a separate study, we have shown that overexpression of outer kinetochore components does not adversely affect kinetochore function (although not specifically *DAD4*, Herrero & Thorpe PLoS Genetics2016). However, to examine this in more detail we have quantified the total cellular levels of Dad4 in an *hmo1∆* strain and find that they are elevated in approximately 30% of cells. However, nearly half of *hmo1∆* cells have normal Dad4-YFP levels (quantitation provided in Figure 5—figure supplement 1), therefore changes in Dad4 levels are not sufficient to explain the aberrant Dad4 foci shown in most *hmo1∆* cells in Figure 5(also in text, in the aforementioned paragraph).

*3) The clustering of complexes (Figure 6) is not that convincing. the approach seems like an awkward strategy for obtaining information that is much more effectively obtained by mass-spec. For example, the histones fall into two distinct categories and the CPC, NDC80, Spc105 and MAPs complexes don't seem to cluster at all. This seeming disparity with known interactions requires explanation.*

We agree that structural information is best achieved using other methods, such as protein-protein interactions (PPIs) identified by, for example, mass spectroscopy. We realize that the SPIs are distinct from PPIs and have attempted to clarify this, see point 17 below. In our original submission we gave the impression that SPIs predict structure which is misleading, rather SPIs identify proteins which behave similarly when associated with specific other proteins, in some cases reflecting characterized functional complexes. We think that the power of the SPI technology is as stated in the Discussion:

“These data illustrate that SPIs can be used, like physical interactions, to reveal the functional organization of the cell. However, since the readout of SPIs is phenotypic, in this case cell growth, the SPIs indicate functional interactions rather than physical interactions per se.”

Some structural complexes are inferred from the SPI data, however this is likely via a common functional association and the SPI data cannot be used as structural evidence. We have now made clear that the SPIs do not constitute structural data – this is discussed more in point 17 below. Nevertheless, we feel that the cluster visualization allows readers to clearly see that SPI data for certain groups of proteins are more similar than for other groups. We have simplified Figure 6 to only highlight 3 clusters (CCAN, DAM1 and KMN). As discussed in point 17 and 18 below we now speculate on why structural complexes may not cluster together using SPI data and conclude the Results section as follows:

“Hence, although SPIs do not substitute for physical interaction data they indicate a common phenotype produced by specific protein-protein associations.”

*4) Results, second paragraph: “we found that 98% of GBP-GFP combinations […] do not affect the growth of cells.” Do 98% not affect growth at all, or are they using a z-score threshold? If z-score, this should be clarified, as it is not necessarily the same as "no effect on growth".*

We do not use a defined z-score or log growth ratio (LGR) as a cutoff for interactions. One reason for not using a z-score cutoff is that although within a screen the LGR between experiment and controls shows a linear correlation with z-score, this is not true between screens as z-scores were calculated on the data from each screen individually, not by combining the LGRs from the whole dataset. Thus a z-score in one screen does not equate to a z-score in another screen. We could have used an LGR cutoff, however, this assumes that SPIs with one target protein would produce a comparable growth phenotype with SPIs from a different target protein, which is an assumption we were not comfortable with. We felt that the best way to confirm SPIs was to sequentially retest the strongest interactions (as judged by z-score) from every screen. Thus the strongest SPIs were retested with 16 replicates and any target-query interaction that gave an LGR that was consistently more than control strains (a non-GFP strain) was considered a SPI. We stopped this retest process when the false discovery rate reached 40% (Figure 4—figure supplement 1). It is rare for such a large screen to retest all of the ‘hits’, but we felt that this was the highest quality approach to allow us to produce a list of repeatable interactions that lead to a growth defect. All of the LGRs from the retests and original z-scores are reported in Figure 4—figure supplement 1. This is now discussed in the text (Results, fifth paragraph). We also note that some SPIs are likely missed due to the colony measurement methodology and furthermore we did not retest beyond a 40% false discovery rate.

*5) Results, third paragraph and Figure 2: “Fluorescent imaging confirmed that ~83% of interactions do occur and typically result in protein relocation […] Of the 524 GBP-GFP combinations that we could score, 435 were detectably colocalized (Figure 2), indicating that in most cases the protein-protein fusions do occur.” Figure 2 shows this and 2D breaks it down in more detail. However, ~50% of the colocalized strains belong to the "indistinguishable" category = proteins that normally colocalize (2D). It is misleading to include this category for estimating how often relocation/interaction occurs. Correcting for this will result in a percentage dramatically lower than 83%. Similarly, this should be clarified in the counts in the text ("435 of 524") and the Figure 2 legend.*

We have amended the 83% figure to 72% by excluding the 40% of samples that were already colocalized, the sentence now reads:

“In cases where fluorescent imaging was able to detect protein relocalization, we confirmed that ~72% of interactions do occur.”

This is now re-iterated in the legend of Figure 2 (and Results, third paragraph) that 40% of the strains examined have GFP and GBP in the same cellular compartment, such that their colocalization status cannot be assessed with microscopy. We have amended Figure 2 to remove the indistinguishable (and uncharacterized) category and clarified this in the legend. The Figure 2 legend now indicates that 210 of the 552 combinations examined already colocalized;

“[…] note that the colocalized category includes 210 combinations where the target and query proteins are within the same compartment and so protein-protein association will not be apparent from this microscopy analysis.”

6) Results, fifth paragraph: “Around 2% of the forced interactions restrict growth […]” Specify z-score cutoff used to qualify as restriction in main text.

We have now clarified this as described in point 4 above.

7) Results, fifth paragraph: “[…]of the 727 SPI proteins […]” -> of the 727 SPI query proteins.

We have included this specific change and also several other instances of ‘SPI proteins’ throughout the manuscript.

8) Results, fifth paragraph: “[…] whose sequestration to another compartment is lethal […]” Did they show it's actually lethal or just decreases growth (i.e. sick)?

We were wrong to state ‘lethal’ when the data show a growth defect (i.e. sick). We have amended the text (Results, end of sixth paragraph).

*9) Results, fifth paragraph: Discuss why there is a difference in suppression of interactions between the frequent and non-frequent SPI groups.*

This issue relates to point 8 also. We have moved the section concerning suppression of frequent SPI query proteins up to follow on from the characterization of the SPI query proteins (Results, sixth paragraph). We have expanded the explanation which now concludes as follows:

“Thus we conclude that these frequent SPI query proteins are predominantly those whose essential function is location-dependent and whose sequestration to another compartment results in a growth defect (as is routinely achieved using other systems (Haruki et al., *2008*)).”

*10) Results, sixth paragraph and Figure 4: It's here stated that target proteins from the same cellular compartment give similar SPIs. While the figure and analysis are suggestive of this, it would be better to carry out a statistical test to corroborate the statement. E.g. Box plots of the distributions of SPIs from same vs. different compartments, accompanied by a Wilcoxon rank-sum test to determine significance.*

The relationship between compartments and z-scores is interesting. We find that in some cases, z-scores correlate between two screens in which the target proteins were from the same compartment, for example Pus1 and Rad52 target proteins, which both localize to the nucleus (Figure 4). We now include a section to describe this compartmentalization in more detail. 7.1% of the possible associations are between a target and query protein within the same compartment, however 10.4% of the SPIs occur between members of the same compartment. This enrichment is statistically significant using Fishers exact test, but nevertheless quite a small enrichment. By including the figures, readers can get a clear picture of the enrichment (Results, seventh paragraph). We give an example of Nop10 (a nucleolar protein), which has an enrichment for SPIs with query proteins in the nucleolus and use Fisher’s exact test to calculate a *p*-value, but importantly, we note that this is not true for all compartments.

In contrast to the number of SPIs within a compartment, when we compare the log growth ratios (i.e. the degree of growth restriction) of strains in which the query and target proteins are from the same compartment, the growth restriction is less than that for associations between different compartments. We have now included a description of this in the text, see Results, seventh paragraph and also included the box plot of same vs. different compartment LGRs in Figure 4, accompanied by a Wilcoxon’s rank-sum test. In summary, there are (slightly) more SPIs between proteins in the same compartment, but the SPIs between proteins in different compartments produce a stronger growth defect.

*11) Results, seventh paragraph and Figure 4: The authors are comparing two distributions of PPI counts (SPI vs. non-SPI) and compute a p-value for the difference using Spearman's rank correlation. I doubt Spearman's rank correlation can be used to produce a p-value for the difference between two distributions. Additionally, the stated p-value (2.2E-16) appears extremely optimistic given the large overlap of the box plots.*

We apologize for using the wrong statistical test, furthermore the *p* value should have been <2.2E-16 (the limit provided by the R function we had used). The *p* value of the Wilcoxon rank-sum test is similarly small. We failed to specify in the figure legend that the top of the box plot was cut off, hence some outliers were excluded. We have changed the y axis of this plot to a Log scale to allow all the data to be included and changed the box plots to include notches that indicate 95% confidence intervals of the median (Figure 4), with a full description in the legend. This makes it easier for readers to see and interpret the data.

*12) Results, seventh paragraph and Figure 4: The CLIK analysis should be described better.* We have now described the CLIK analysis in more detail (Results, seventh paragraph).

*13) Results, eighth paragraph: Describe or reference the gene ontology enrichment analysis.*

We have included a reference for the GOrilla algorithm for ontology enrichment.

*14) Results, eighth paragraph: Was fluorescently tagged Nuf2 examined in these cells as well? How is Nuf2 affected?*

To characterize the effect of Hmo1 on the kinetochore in more detail, we have now created ten additional kinetochore proteins fused with GBP (two of which are tagged at both the N- and C- termini) and tested the effect of introducing these into an Hmo1-GFP strain; we have now discussed this in the text (Results, eighth paragraph) and in a new Figure 5. This experiment shows that the effect of Hmo1 association is not restricted to Nuf2, but is also found for Mif2 and Ctf19, a central and outer kinetochore component respectively; although a number of other associations do not produce a phenotype. We chose to examine Dad4 and Mtw1 as they are canonical members of the kinetochore complex and both have previously been quantitatively characterized in some detail (published work from both our own lab and Kerry Bloom’s lab). This rationale is now discussed in the text (in the aforementioned paragraph).

*15) Figure 5: Explain why Mtw1 was chosen.*

See point 14 above.

*16) Figure 5: Describe in much more detail how to interpret these plots. Also, what are the error bars? Finally, is the p-value (E-10) really correct? Depending on what the error bars represent this looks low…*

We have amended the Figure 5 legends (now C & D) to explain these plots in more detail. The barcharts showed means +/- standard deviations. However, we have changed these plots to notched box and whisker plots and given a full explanation in the legend. We have changed the *p*-values from two-tailed unpaired *t*-tests to Wilcoxon’s rank-sum tests – comparing the foci intensities of WT kinetochore foci with those of the mutants (for *hmo1∆* vs WT the *p*-value is 4x10^-24^ and for *sgf29∆* the value is 4x10^-37^). The distribution of foci fluorescence intensities (right panel), particularly for *hmo1∆* cells, is not a normal/Gaussian one, hence we wanted to show the distribution of the fluorescence data so that readers can get a clearer picture of these data (the right panels in Figure 5).

17) Results, last paragraph: “[…] SPI data can be used to predict protein complexes […]” Substantiate this claim. Show if the SPI data actually have predictive power with a ROC or precision recall curve.

This point follows on from point 3, we do not wish to claim the SPIs substitute for PPIs. We realize that this was an over-statement of our data to say that SPI data can predict protein complexes or would substitute for protein-protein interaction data, since SPI data is not structural. Rather, we want to indicate that when specific proteins are associated with the same protein complex they may produce a similar SPI phenotype. We have changed this section to

“It is important to note that SPIs are not a substitute for physical interaction data, but rather represent a common phenotype in response to forced association. […] Thus although the proteome-wide SPI data themselves do not directly give structural information, the SPI data does group query proteins within these known protein complexes.”

We have also amended the quaternary structure section to reflect this;

“We nest asked whether the SPI data would correlate with the quaternary structure of multi-protein complexes, since protein associations with one part of a complex may give a similar growth phenotype that contrasts with a different part of that same complex.”

This section now fits with our speculation of the different phenotypes given for members of the DAM1 complex (Results, last paragraph, as requested in point 18 below). We speculate here why some protein complexes may not cluster together using SPI data.

Furthermore we conclude the Results section with the sentence

“Hence, although SPIs do not substitute for physical interaction data they indicate a common phenotype produced by specific protein-protein associations”.

*18) Results, last paragraph: Try to interpret why the DAM/DASH complex segregates into distinct clusters.*

We have expanded this section of the text to speculate on why the DAM/DASH complex splits into two clusters based upon high-resolution structural models of this complex, see Results, last paragraph. This addresses issues raised in point 3 and 17.

*19) Discussion: I suggest explicitly writing "physical interactions" instead of "interactions" to not confuse with SPIs.*

We have changed these in the Discussion.

*20) Discussion: “[…] and derive quaternary structure […]” ‘provide information on’ would be more appropriate than ‘derive’.*

We have changed this in the Discussion.

21) Discussion: “[…] 54% (394) are conserved in human cells.” May be worth discussing how this compare to the conservation of the complete yeast genome to human?

We have amended these values utilizing data from ‘Yeastmine’ (yeastmine.yeastgenome.org). 3766 out of 6604 yeast ORFs have human homologs (57%), 549 out of the 727 SPIs have human homologs (~76%) (Fisher’s exact test, *p*-value = 3.32x10^-23^).

[Editors' note: further revisions were requested prior to acceptance, as described below.]

Reviewer #1: The authors have largely addressed the concerns I raised and put in appropriate caveats in their revised manuscript. I am still concerned that their Abstract is misleading in claiming to establish that proteins have an "unanticipated tolerance for forced protein associations and consequently their relocation". I think it is likely that in many, perhaps the large majority, of the cases where apparent relocation of a protein does not disrupt function has to do with at least partial retention of the protein in its correct locations. It would indeed be quite surprising if most nuclear proteins could function in the cytosol or most organellar localized proteins could function outside of their native organelle. I do not think this is what the authors intend to say (and certainly is not what they have shown) but I could easily see how a casual reader of the Abstract could be left with this impression.

Reviewer #1 had asked for one change to the Abstract, we have changed:

“This analysis reveals that cells have a remarkable and unanticipated tolerance for forced protein associations and consequently their relocation.”

to

“This analysis reveals that cells have a remarkable and unanticipated tolerance for forced protein associations, even if these associations lead to a proportion of the protein moving compartments within the cell.”